# Weaving a cocoon on the way to aging transcendence: Grounded theory study on aging perception during menopause transition

**Shahin-Dokht Navvabi-Rigi**[1,2], **Farahnaz Mohammadi-Shahboulaghi**[3,4]*,
**Mahshid Foroughan**[1,4], **Yadollah Abolfathi Momtaz**[1,5]

**1** Department of Aging, University of Social Welfare and Rehabilitation Sciences, Tehran, Iran, **2** Pregnancy Health Research Center, Zahedan University of Medical Science, Zahedan, Iran, **3** Department of Nursing, University of Social Welfare and Rehabilitation Sciences, Tehran, Iran, **4** Iranian Research Center on Aging, University of Social Welfare and Rehabilitation Sciences, Tehran, Iran, **5** Malaysian Research Institute on Ageing (MyAgeing), University Putra Malaysia, Serdang, Selangor, Malaysia

☯ These authors contributed equally to this work.
* mohammadifarahnaz@gmail.com

**Data Availability Statement:** All relevant data are within the paper and its Supporting Information files.

## Abstract

During menopause, women experience major changes, including the onset of aging as a natural and inevitable event. The present study aimed to explore and explain the process of aging perception during menopausal transition. This study was a qualitative grounded theory research, which was going to be implemented in 2019–2020 on 18 Baluch women settled in the Sistan & Baluchestan province in South-East Iran and among midlife or older menopause experienced women. Sampling was started first by Purposive sampling, and then it was performed with theoretic sampling. Data analysis was performed according to Corbin and Straus's approach (2015) in four phases: (1) identifying and developing concepts; (2) analyzing data for the context; (3) entering the process stage into analysis; and (4) integrating categories to build a theory. In this study, seven main categories were obtained: "Sunset of youth", "aging as the other side of the coin of menopause", "Weaving a cocoon", "aging as a mental trap", "social acceptance", "aging domino", and "feeling of transcendence". It seems that menopause plays an important role in Baluch women's view toward aging. The practical results of this study can be applied to better understand the middle-aged and older Baluch women's attitudes toward aging. Also study shows a new evolutionary and situational perspective on the lives of middle-aged menopausal Baluch women. Baluch women in menopause accept the sunset of youth and look forward to experiencing the Feeling of transcendence. Identify and respond to their needs by developing and establishing health policies to change their negative attitudes.

**Funding:** I confirm that we have not received any funding to support this work.

**Competing interests:** No potential conflict of interest was reported by the authors.

## Introduction

Aging is a natural process in an individual's life. Changes associated with old age can form a new self-perception in an individual, known as "aging perception". The aging perception is a measure of an individual's satisfaction with the process of aging, reflecting one's adaptation to age-related changes. This perception depends on the persons' understanding of the aging process in their sociocultural context [1]. It seems that exploration of middle-aged and elderly people's understanding of this period can be a way to understand the aging process and people's experiences of transition to old age [2].

The life of every individual consists of different periods [3]. Generally, the life cycle is considered as a socio behavioral concept, influenced by age, time, personal development, and life stages, events, and transitions. Different transitions in life often begin with experience of loss [4] or changes in important life roles, shaped by one's perception [5]. Menopause is one of these transitional events, which is considered to be an important turning point in a woman's life and health [6] By 2050, a major global challenge is speculated to be the feminization of the elderly population [7].

During menopause, women experience major changes, including the onset of aging as a natural and inevitable event [8, 9]. Nearly one-third of a woman's life is spent during menopause [8]. It is associated with a shift from maximum active fertility to infertility and finally a transition to later stages of life [10]. In the literature, it has been discussed as a biopsychosocial indicator of the female aging process [11]. Research shows that the onset of aging is relatively marked by menopause in women [12, 13] therefore, understanding women's perception of aging during menopause is very critical [12, 14].

Delyser [15]and Lockenhoff [16] reported a review of attitudinal research on menopause and aging in different communities shows positive, negative, and neutral attitudes toward menopause. Research also showed that menopause and aging were defined differently in different cultures, and Some cultures consider menopause as a taboo and relate it to aging; Women are regarded as a source of fertility, and menopause as an indicator of fertility termination [16, 17] aging, and the end of women's beauty; therefore, it has negative connotations [18]. In contrast, some studies have reported an association between menopause and aging with women's increasing wisdom, knowledge, and respect [16, 17]. However, there are also studies reported an indifference toward menopause in societies [17] or regarding it as a hidden experience, based on the cultural context [13].

According to the 2016 census, 7.4 million elderly people live in Iran, 50.6% of them are females [19]. Iran is home to many ethnic groups, including Turks, Kurds, Lors, and Baluch people, each with its special cultural norms. The aging experience can be considered different in different cultures and among different ethnicities. One of the ethnic groups live in Iran is Baluch community, who are settled in one of the largest provinces in the southeast of Iran [20]. This province is a less developed and deprived one, so Baluch people are struggling with many difficulties such as poverty, low sanitation, water scarcity, and bad climactic conditions. Baluch women have a difficult life and face many challenges, such as younger menopausal age relative to the world's average (47.3 to 51 years) [21], higher fertility ratio relative to the country's average [19], increased maternal mortality [22], low individual independence [23], low out-of-home employment, and higher vulnerability to stressful events as compared to men [24].

Living in a patriarchal community, Baluch women face numerous gender-related problems that make their journey through life cycles complicated. Menopause as a transitional stage from middle age to aging may have special impact on the aging perception among women. As mentioned earlier, the aging perception is influenced by the person's mentality based on

cultural context [25]; therefore, investigation on aging perception, as a culture bound phenomenon, with an emphasis on menopause, can help us to extend our knowledge about the diversity of aging experience among the women over the globe.

Most previous studies on the relationship between menopause and aging have employed quantitative approaches, which cannot explain the nature of the process. Few qualitative studies, while highlighting the importance of examining the nature of the phenomenon, used the content analysis approach [12, 18, 26, 27] or phenomenology [13, 28, 29]; however, they did not explain the process of aging perception during menopausal period. Accordingly, the present study aimed to explore and explain the process of aging perception during menopausal transition in Baluch women.

## Materials and methods

### Design & participants

This qualitative study aimed to construct theory grounded in data. The method used in this study reflects Corbin and Straus's approach to grounded theory analysis [30]. The study assumptions were derived from pragmatist and interactionist philosophies. The participants of this study included 18 Baluch women at different menopausal stages during 2019–2020. The first step Purposive Sampling was performed based on inclusion criteria with maximum diversity in sampling (marital status, having children or not, different menopause ages, socioeconomic status) by researcher in the comprehensive health service centers, in Zahedan in the southeast of Iran and contacted the participants and conducted interviews at the center or at home, at their request. In the next step, theoretical sampling was performed, and based on received answers. The main concepts were formed, and the relationships between these concepts were explained.

The inclusion criteria were as follows: Iranian Baluch women who could understand/speak Persian and have logical verbal communications (to ensure the adequacy of cognitive ability); experiencing menopause(complete cessation of menstruation for one year or more) or in perimenopausal stage (irregular menstruation/cessation of menstruation for at least 60 days or more); or having clinical symptoms of menopause (e.g., hot flashes, sweat, and insomnia in at least the past six months), as approved by a physician or a midwife. The exclusion criteria were as follows: women with menopause due to surgery (hysterectomy or oophorectomy), medication use, chemotherapy, and having a psychiatric diagnosis.

### Data collection

In-depth semi-structured, face to face, individual interviews were conducted to collect the data until reaching data saturation (no new codes were formed) The interviews were conducted in comprehensive community health centers or in women's place of residence (participants' preferences) and each lasted about 30–60 minutes.

The first author (a native female researcher trained in qualitative method with clinical experience in Baluch community) began the interviews by introducing herself, clarifying the goals of the study, and obtaining informed consent. Some of the main interview open- ended questions were as follows: "Can you explain your menopause and reproductive aging?", "How did you find out you were going through menopause?", "How did you feel when you find out it?", "How did you your life changed during this time (in family and among other people)?", and "What do you think has changed?"

The main questions were followed-up by "why", "when", "what", "where", and "how" questions. If the participant referred to aging when describing her aging experiences, it would be explored by follow-up exploratory questions. Most of the interviews with female participants

were conducted individually and recorded by a cell phone. After listening to the recordings several times, they were transcribed in Persian.

## Data analysis

Data analysis was performed according to Corbin and Straus's approach (2015) in four phases: (1) identifying and developing concepts; (2) analyzing data for the context; (3) entering the process stage into analysis; and (4) integrating categories to build a theory. The statistic methods used in order to analyze and identify the data which were the context, process and consequences based on paradigm analytic tools such as asking questions from the data (sensitive questions, theoretical questions, Questions that were more practical in nature and guiding questions), and comparing (constant comparison and theoretical comparison). In this study, each interview was recorded, typed and transcribed, and initial code and then initial concept by researcher identified. In analyzing the data for context, concepts linked in order to establish between categories, and linked them together in a new concept through ongoing comparison. In the stage of process of the analysis, the relationships between intra-categories systematically and integrating the categories (subject labels based on integrated memo titles used to name concepts). In the stage of integrating the categories, the study's storyline, core concept, hypothesis emerged by way of the relationship formed between the categories and clarification. In this Statistical analysis was not used any analytical app and it was done manually. Moreover, the criteria of credibility, dependability, conformability, and transferability, approved by Corbin, were used to meet the acceptability/validity criteria. The analysis was done by researcher and it was evaluated by two co-worker who were an expert in qualitative research.

To increase acceptability, a combination of semi-structured interviews with note-taking at the scene and reminders was used along with other strategies, such as long-term engagement with the data and spending enough time on collecting, analyzing, and reviewing the data. Although the researcher was immersed in the subject under study for months, the supervisors, consultants, and participants' opinions and feedbacks were also considered [31]. Moreover, to meet the criteria for data transferability, validity, and robustness, the researcher tried to make it possible for other scholars to follow the research path through in-depth description and analysis of the participants' background and characteristics, description of the study context, and clear description of obstacles and limitations. A comparison of the present findings with other studies was also made to assess the appropriateness of the collected data. A total of 22 interviews were conducted, four of which were supplementary interviews (Table 1).

## Ethical consideration

The study was approved by the Research Ethics Committee at the Welfare and Rehabilitation Sciences University in Iran (Ethical code: IR.USWR.1398.26). The participants were informed about the study verbally and were assured of confidentiality and anonymity. They were informed that they could withdraw from the study at any time. The participants provided written informed consent before the interviews.

## Results

In the present study, 2311 primary codes were obtained, yielding in 14 subcategories after comparing similarities and differences in the analysis stage. A total of seven main categories were obtained after continuous and theoretical comparisons of subcategories. According to the paradigm analytic tool, the results showed that during the event of menopause, women based on the conditions (Sunset of youth, aging as the other side of the coin of menopause), chose the strategy of weaving a cocoon and waiting within it for the consequences (aging as a

**Table 1. Characteristics' demographic of participants.**

| participant | Age | Menopause age | Education level | Marital status | Occupation status |
|---|---|---|---|---|---|
| 1 | 50 | 48 | Middle literacy | Married | housewife |
| 2 | 57 | 52 | illiterate | Widow | housewife |
| 3 | 54 | 43 | illiterate | Married | housewife |
| 4 | 67 | 47 | illiterate | Widow | housewife |
| 5 | 41 | 35 | Bachelor | Widow | Government employee |
| 6 | 65 | 40 | illiterate | Married | housewife |
| 7 | 62 | 49 | elementary | Married | housewife |
| 8 | 70 | 49 | illiterate | Widow | housewife |
| 9 | 60 | 49 | Middle literacy | Married | housewife |
| 10 | 54 | 46 | elementary | Married | self-employment |
| 11 | 55 | 47 | elementary | divorced | self-employment |
| 12 | 100 | 50 | illiterate | Widow | housewife |
| 13 | 56 | 47 | Middle literacy | Married | housewife |
| 14 | 41 | 40.5 | Diploma | divorced | Government employee |
| 15 | 42 | 38 | illiterate | Widow | housewife |
| 16 | 68 | 35 | elementary | Married | housewife |
| 17 | 50 | 46 | Middle literacy | Unmarried | self-employment |
| 18 | 49 | 42 | Diploma | Married | self-employment |

mental trap, social acceptance, aging domino, and feeling of transcendence) to appear (Table 2).

## 1. Sunset of youth

The sunset of youth is one of the main categories of the present study and a part of the structural and contextual conditions of the aging perception process in Baluch women. Advancing age or waning youth was the main concern stated by the participants. As soon as women faced menopause, they started to experience a feeling of becoming old and threatening by the aging challenges, such as Physical, sexual, and psychological declines. Therefore, they were entering a fearful stage, waited each month for menstruation to come and by losing their hope, they gradually started to feel that they are old. This main category included the subcategories of "fear of aging and hope for rejuvenation", "threats of aging", "physical manifestations of aging", and "psychological manifestations of aging".

**Fear of aging and hope for rejuvenation.** his subcategory referred to a feeling of being old in the women due to the possible challenges brought about by menopause, suggesting the importance of the menstrual cycles in the lives of these women. Despite the emergence of menopausal symptoms, Baluch women waited for the occurrence of menstruation each month and were focused on their menstrual signs and symptoms in the absence of bleeding. At first, women hoped that these changes are not related to menopause, but later, they were inevitably faced with the truth. Most of the participants stated that they were preoccupied with menstruation before menopause. In this regard, one of the participants said:

> "...When it was the time, my body, my back, and my abdomen ached; I got heavy and felt down...I felt like this for 4–5 days a month for several years. I was preoccupied with it." (p. 9, 60 years old)

**Table 2. Process of aging perception in menopause.**

| Initial categories | Subcategories | Main categories | | Core category |
|---|---|---|---|---|
| • feeling of premenstrual syndrome (PMS)<br>• preoccupied Waiting for menstruation | Fear and hope waiting for rejuvenation | Sunset of youth | **Conditions** | **Weaving A Cocoon On The Way To Aging Transcendence** |
| • "feeling of Drought"<br>• feeling of advancing age<br>• Feeling of worn out<br>• Loss of spirit | Threats of aging | | | |
| • Gradual apparent decline<br>• Onset of pain and physical weakness<br>• Gradual physical decline<br>• Gradual onset of ability decline | Physical manifestations of aging | | | |
| • Amnesia<br>• Distractions<br>• Incidence of anxiety<br>• Occurrence of mood changes | Psychological manifestations of aging | | | |
| • The culture of menopause secrecy<br>• ،Belief in the loss of fertility value<br>• ,"Menopause is a defect /disease"<br>• Negative attitudes about the association of menopause with aging "bad aging" | Cultural importance of fertility | Aging as the other side of the coin of menopause | | |
| • Experiencing the consequences of menopause in others Stress of family reactions to menopause- | Menopause-related environmental stress | | | |
| • Menopause secrecy<br>• Change in the appearance of clothes | Changes in self-concept | Weaving a cocoon | **Action/ interaction** | |
| • Reduce interpersonal communication<br>• Inattention in self-care | Reduced level of activity | | | |
| • Avoiding your spouse<br>• Suppression of sexual desires- | End of marital relation | | | |
| • Calm yourself with the light of spirituality<br>• ،Distraction of thought from the feeling of aging | Acceptance of one's fate | | | |
| • A period of confusion<br>• Uncertainty distances<br>• Fluctuation of feelings about social status | | aging as a mental trap | **Consequences** | |
| • Support and respect from the community<br>• Acceptance of menopause by spouse | | social acceptance | | |
| • Loss of courage<br>• Deficiency of self-efficacy<br>• Feeling of Disability | Sunset of youthful agility | aging domino | | |
| • Feeling depressed<br>• Gerascophobia<br>• Self-ugliness | Insecurity and boredom | | | |
| • Feeling of rationality<br>• Adopt the role of consultant | Feeling of superiority | feeling of transcendence | | |
| • Feeling pure""<br>• A sense of closeness to God | Spiritual maturity | | | |

"*Women who bear children at the age of 50 or 60 years are still considered young. I saw someone on TV who gave birth to a child at this age.*" (p. 16, 62 years old)

**Threats of aging.** Women believed that they were young as long as they had their menstrual periods. They considered menopause as the end of fertility and called it "Drought out" due to the emergent weakness and dryness of female reproductive system. They talked about it with sadness and regret and asked about its nature and inappropriate timing:

"*I got completely dried out. I think I got old; and it was over.*" (p. 16, 62 years old)

This participant also stated:

"*The end of menstruation is related to aging and becoming dry. As we age, the uterus gets drier, and the dirty blood is dried in it; this is aging.*" (p. 16, 62 years old)

Moreover, regarding the "non-excretion of dirty blood" during menopause, one of the participants said:

"*I think when I menstruate, my body becomes lighter, and my eyesight improves. We think that during menstruation, the blood that comes out is poisonous by itself; I want this poison to leave my body. Although I am in so much pain during menstruation, I am willing to take it, without a single Hyoscine tablet, like my sisters and many other women in my family, because we believe that this poison must leave the body. . .*" (p. 14, 41 years old)

**Physical manifestations of aging.** The participants believed that the accumulation of dirty blood causes negative physical signs and symptoms:

"*Immediately after menopause, our eyesight is reduced, our hands and legs feel painful, and we get headaches. . .We start to have a poor vision. . .*" (p. 6, 65 years old)

**Psychological manifestations of aging.** The Baluch women's perspective, with the loss of ovarian function, other organs will also undergo changes. It even affects the individual's activities and psychological status (lower mood, anxiety, etc.). In this regard, one of the participants said:

"*For this reason, I feel old. . .I experience mood swings constantly since menopause; I became nervous; I am not who I was before.*" (p. 9, 60 years old)

## 2. Aging as the other side of the coin of menopause

It is one of the main categories of the current study and one of the structural and contextual requirements for understanding the process of aging perception in Baluch women. This category refers to the existence of a set of prevailing beliefs, attitudes, and opinions in the Baluch community regarding menopause, which is rooted in their cultural context and reminds people of the aging process. This main category included the following subcategories: "cultural importance of fertility" and "menopause-related environmental stress".

**Cultural value of fertility.** Due to the cultural importance of fertility in Baluch people (having more children is equal to being more valuable), women do not discuss of their menopause and hide it as long as possible. These women attributed this secrecy between infertility and being old. One of the participants said:

"*I had a party two nights ago. I asked someone about how many years had passed since her menopause; she told me that she was not so old and that she was still young! This is how it is; if we tell the truth, people say that we are old, or our husbands think that we cannot have any more children; so, they leave us and remarry.*" (p. 3, 54 years old)

Baluch women also considered infertility to be synonymous with disease "Menopause is a disease":

"*Because your body is too disturbed in every possible way, you feel that you are not alive anymore, you just breathe! Now, I have severe arthritis; I have pain in my legs, severe pain; I did not have it before, I became like this after menopause; I feel pain in all my bones; I have a heart disease; I feel very dizzy, and my eyes become blurred...*"

Some women considered menopause as a form of "bad aging":

"*We (women) do not have a good aging; it is very difficult. We become discouraged and disappointed with the world. Our weaknesses lead to sickness; we may go blind, or we may go deaf. Things have gotten worse since my menopause; now, I am completely worn out.*" (p. 13, 56 years old)

**Menopause-related environmental stress.**   The menopausal experiences of family members and relatives stated as an influential factor in creating a feeling of becoming old by the participants. Regarding the negative consequences of menopause in her mother, one of the participants said:

"*My mother had too much pain in her legs and her lower back; she had many problems. She used to say that when menopause starts, all diseases attack on your body, and you become powerless.*" (p. 3, 54 years old)

### 3. Weaving a cocoon

It is one of the main categories, indicating the interaction between the aging perception and menopausal transition among women. In the Baluch community, women do not directly discuss of their menopause. Instead, they express it in covert ways such as changing their lifestyle and clothing; they wear darker colors, with simple designs, and limit the level of their activities. A woman's alienation from the society begins by avoiding her partner. Because she feels old and unattractive, she refuses to have sex with her husband; therefore, she suppresses her sexual desire to show her reaction to her aging. This category consisted of the following subcategories: "changes in self-concept", "reduced level of activity", "ending the sexual relation", and "acceptance of one's fate".

**Changes in self-concept.**   This subcategory implied that Baluch women indirectly expose their menstrual transition by changing their image in the society and for themselves. One of the participants stated:

"*This dress that I am wearing right now, although it has decorated by small flowers, not big ones, is not suitable for me; it does not suit my age. Its color should be darker, because I think I've gotten old...People believe that it's unsuitable to wear some clothes as you get older, because you seem too flashy; the darker your clothes are, the more respectable you look...*" (p. 1, 50 years old)

Moreover, about hiding menopause, one of the participants said:

"*When my menstruation stopped, I did not tell my husband or anyone else. There was no need to say it; we did not have sex anymore, and we slept in separate beds. He also did not know if I was menstruating or not.*" (p. 4, 67 years old)

**Reduced level of activity.**   This subcategory implied that women somehow deliberately reduced the level of their activities. One of the participants said:

"*Sometimes, I go for a walk; I walk a little and then I have to sit down again. The cars stop by and offer me a ride as if I cannot walk on my own; I really hate this. They think I am miserable, poor, and lonely and have no one; that's why I am not walking alone*! *I hate walking the streets like this.*" (p.3,54 years)

**End of sexual relation.** This subcategory involved women's suppression of their sexual desires. In this regard, a participant said:

"*We used to have sex for a while, but not anymore; I mean, I have become more masculine. I have become someone else; at first, I wanted him to come to me, but not anymore; I did not care about our marital relations at all.*" (p. 9, 60 years old)

**Acceptance of one's fate.** The participant women turned to religion to distract and calm themselves. In this regard, one of the participants said:

"*The positive thing about menopause is that you can pray as much as you want. You can go to Quran sessions and touch the Quran (in Islam, it is a sin to touch the holy Quran during menstruation). When menstruating, you cannot do these things; you can only recite God's name, but now, if you do not pray, you will regret it.*" (p. 9, 60 years old)

## 4. Aging as a mental trap

It is one of the main categories, related to the women's reaction to aging process, and refers to feelings of confusion and uncertainty, which come along with changes in their social status. Permanent cessation of menstruation is the end of a stage in a woman's life, and menopause is the beginning of a new phase in her body. So that with the onset of premenstrual symptoms each month, they suspected of still being young, and when the symptoms were over, they felt old again. These cycles were repeated even for several years, creating a feeling of entrapment in the women and the image that they have reached to the end of youth and "There is no way out", for them.

"*At first, I did not notice it; of course, I suspected it as my body was in pain. When my menstruation stopped, I asked myself what is happening to me*?" (p. 8, 70 years old)

"*I am both middle-aged and old*! *I mean I think so.*" (p. 9, 60 years old)

Regarding the feeling of change in status, one of the participants stated:

"*They say that I'm always sitting and that I always fall behind things; I hate it. Some women get old at the age of 50 years, while some like me start to age from 37, women who have worked hard and faced many problems, tolerating their husbands' constant nagging. In the end, the husband and children will blame you for it.*" (p. 13, 56 years old)

## 5. Social acceptance

It is one of the main categories, suggests that family support and respect can lead to acceptance of menopause by the women during transition from middle age to old age. The supportive behavior of the family and specially the spouse, which mostly occur over time, can lead to the acceptance of menopausal women by themselves and by the society. Regarding family support and respect, one of the participants stated:

"*Thank God I have a child at home who does the house works. I felt confident at that time; I did not feel depressed at home at all.*" (p.3, 54years)

Finally, the spouse's support was introduced as a determinant factor:

"*My husband told me that he did not know what had happened to me; my poor husband did not tell me anything about it anymore…*" (p.10, 54 years)

## 6. Aging domino

This main category indicates women's feelings about their physical, functional, and psychological declines, causing them to socially isolate themselves. This category included the two subcategories of "Sunset of youth agility" and "insecurity and boredom".

**Sunset of youth agility.**    Before transition to menopausal and experiencing the process of aging, women are interested to do difficult activities, as they feel proud of their energy of youth and fearlessness. With the onset of aging and menopause, they feel that their physical strength starts to decline. So feeling frustrated, they lose their courage to do many of their previous activities:

"*…When we are young, we feel proud of ourselves, and do not feel hardships of life…When we get older, we avoid everything; we get closer to God because we are weaker and have less power. I feel like this…Yes, we lose our sense of pride.*" (p. 3, 54 years)

**Insecurity and boredom.**    The Baluch women reported that they are feeling boredom after menopause:

"*After the end of menstruation, I felt worse day by day. As you get older, you get depressed; I do not know what you call it. You get Alzheimer's disease; I got Alzheimer's. I swear to God I cannot remember the names of my children. For example, if I want to call my own child, sometimes, I call the names of all my daughters-in-law, first; I call all of them until I can remember the name of my own child.*" (p. 13, 56 years old)

Although some of the participants had many children, they were upset and felt sorry that they could not have children anymore. As if, having the ability to give birth to another child made them more secure. One of the single participants, at the age of transition to menopause, expressed their fears of insecurity due to her possible future loneliness:

"*Maybe those who have husbands and children will not be so affected, but I wonder who will help me; how my future will turn out; or what will happen to me.*"

With the onset of postmenopausal changes, women start to have a more negative view of this transition:

"*Look, my face has wrinkles, and my skin is dark…This started since menopause…*" (p. 12, 100 years old)

## 7. Feeling of transcendence

The Baluch women believed that they could reach high levels of knowledge and rationality after the cessation of menstruation; in other words, these women hoped to reach to the higher

ranks of their society and gradually assume an advisory role in major affairs as they get older, in a way people around them think of them differently. Also, because of their religious beliefs, women who were no longer menstruating felt closer to God, as they could focus on their religious practice. This main category included the subcategories of "feeling of superiority" and "Spiritual maturity".

**Feeling of superiority.** "*Well, now that I am three years older since the start of menopause, my husband respects me more. The longer you live, the more respect you receive from your husband, your children, your family, and everyone else. The older you get, the more valuable you become. Well, aging can be good too; your family will think highly of you. They respect you very much.*" (p. 18, 49 years old)

The other participant said:

"*For example, when my old mother-in-law passed away, it was very difficult for us; it felt as if something very important was missing from our lives.* (p. 18, 49 years old)

**Spiritual maturity.** In the patriarchal society of Baluch, with advancing age, women find the opportunity to reach a social status similar to that of men:

"*It is said that a woman who has stopped menstruating can go to a mosque to pray like a man does. The Baluch people believe in this. For example, my menstruation stopped 10–12 years ago; so, I can stand right behind the Mowlavi (mulla) to pray; I am pure now like a man!*" (p. 11, 55 years old)

### Weaving a cocoon on the way to aging transcendence

The **core category** of this study was "Weaving a cocoon on the way to aging transcendence". The Baluch women, during early phases of menopause and when they were entering a new developmental stage, mostly felt aging as a threat because of their cultural context and society which worshipped fertilization. They were experienced their sunset of youth accompanied by physical, functional, and psychological declines and felt disappointed. Moreover, they stated that they had developed various age-related diseases by the end of fertility and called it "bad aging".

They adopt a secretive approach toward menopause to avoid being labeled by the society; they changed their clothing and are more inclined to stay at home. Since they found themselves vulnerable, they reduced their level of functional activities to prevent further harm. They even avoid sexual relations and somehow tried to end their marital life. As they hided themselves from the society, their inclination toward spirituality increased.

By moving away from worldly preoccupations, women aimed to achieve spiritual growth and stop focusing on their physical, psychological, and functional problems. It seems that women, by losing some abilities, need to acquire a variety of other abilities. They believed that they can get closer to God, using the postmenopausal period to worship God and elevate their spiritual life. Meanwhile, the society, also, accepted them and gave them a higher position due to their developed intellectual and spiritual powers

### Discussion

The results of this study aimed to explore the aging perception of Baluch women during the menopausal transition. The Baluch women in our study remarked that the occurrence of

menopause was associated with a feeling of becoming old, which was accompanied by physical, sexual, functional, and psychological declines in them, emerging in the cultural context of their society. They also felt that despite the decline in their physical strength and the related social stress, during their menopause period, they felt that they are closer to God. They rather consciously tried to avoid many social activities and waited for a time point that they could obtain a high l status as a respected and wise person enable to instruct others in the community. Therefore, the context, actions/ interactions and consequences are discussed in this section.

Exploring the **context** of Baluch society revealed that as soon as menstruation stops, middle-aged Baluch women start to feel older; this suggests the importance of menstruation and fertility in Baluch community. These women believed that with vaginal drying, their eyesight began to dim and this is a sign onset of aging. In their opinion, other organs in their body were also gradually declined, indicating the end of life as a youth and the onset of limitations in the person's movements and many other activities. In this study, women used phrases, such as "getting dried out", to describe the end of the reproductive period.

According to the studies by Ramakuela [13], Araya [12], Yazdkhasti [32], and Ilankoon most women believe that menopause is the onset of aging, also in a study by Yazdkhasti [32], a postmenopausal woman described herself as a "dry yellow leaf" which are somehow similar to the findings of our study. Overall, the aging process in any culture and society may be different and have its own meanings and concepts [13]. The use of such words with negative connotations suggests the threatening aspect of menopause heralding the onset of aging in this population.

In the present study, menopause and aging were assumed to be two sides of the same coin. In Baluch culture, "Menopause is a disease" which means that aging itself is a disease, too. In this study, since most participants described menopause as a disease, the onset of aging which considers as a time for disease was also strongly felt. They believed that the lack of blood excretion led to the accumulation of toxins in the body and resulted in the onset of diseases. On the other hand, lost fertility was considered as a defect by these women and made them avoid discussing it. In a study aging has been compared to disease and death, too [33]. According to a study by Ebrahimzadeh [34], women possibly hide their infertility due to stigmatization. They may need time to disclose their menopausal transition to their husbands. According to a study by Mahadeen [35], menopause is a hidden experience in patriarchal cultures. In studies by Doyal [36] and Sergeant [11], hiding menopause was mentioned as an unspeakable social rule. Other studies [32] suggested that the excretion of impure blood creates a satisfying feeling of purity in women, that in the present study, menopausal women showed that they finally feel satisfied too, because they could obtain an equal social status to men.

One of the most important reasons why women began to hide themselves from the society was the onset of perceived aging, which was associated with menopause and losing their fertility power. In other words, women, by menopause, felt that they gradually lose their beauty and vigor and experienced the signs and symptoms of aging. It seems reasonable as an individual's sense of purpose and meaning in life is influenced by the intrinsic values of the culture that she/he lives in [37], where women sacrifice their needs for the sake of family which is partly consistent with the culture of Baluch. Generally, gender inequality can be seen in different aspects of life in developing countries including sexual life. On the other hand, some studies reported menopause as a positive experience due to the cessation of bleeding and secretions, the reduced risk of pregnancy, and the possibility of participating in religious ceremonies [38]; the last was also seen in the present study.

**Action/interaction** in the present study, "**Weaving a cocoon**" was a major approach in Baluch menopausal woman. The first behavioral changes observed in these women based on

the informal norms of the society, were those which obliged them to behave according to their family role and age; therefore, women made some changes in their appearance including clothing because of both the social expectations and their own feelings of necessity. Similarly, non-verbal signs in other cultures regarding clothing and even jewelry and accessories were found [39]; unlike women in the present study who sought simplicity, women in other cultures tended to look luxurious [39].

After menopause, Baluch women even abstain from sex, because they think that it is not commensurate with their age and status; in this way, they indirectly declare their aging to others. These changes are consistent with the role theory [40]. In this way, Baluch women declare their concerns about the public awareness of their menopause and its stigmatization in the social context. According to the stigma theory, which is the origin of symbolic interaction theory, individuals need to know how others describe them and how they react to them. Therefore, when a postmenopausal woman is distinguished from others and classified as "an aging woman", her self-concept changes, so she decides to hide her menopause to avoid being labeled by the society, especially by the relatives.

During menopause, Baluch women tend to limit their social communications. They believe that when they get older, they should sit at home "modestly" and engage in spiritual practices; this is consistent with the disengagement theory, referring to the elderly's tendency toward isolation [41]. These women limit their physical activities and spend their life waiting for a change in their family roles. Besides, they believe that they can improve their social status through spiritual practices (saying prayers and worshiping God as a kind of meditation), as recommended by the religion in times of crisis [32], This also indicate life course perspective defined as "a sequence of socially defined events and roles that the individual follows over time" [42].

The **consequences** in of the category of "aging as a mental trap" was extracted based on the mindset of Baluch women. When going through menopause, women felt confused and trapped in a dead-end way called aging. On the other hand, some previous research also described the menopausal transition as a new natural stage in a woman's life and reported it as a positive view, unlike the present study. In this study, women described their hope for menstruation each month and its aftermath numbness. However, the time gap, that is, the interval between the onset of menopause and the perception of aging (give up hope), varied between individuals from one year to several years.

The **consequence** in the category of "aging domino" implied that despite still having high physical strength, women did not dare to do many activities with the aim to prevent physical injuries because they felt old and disable. They thought that they are vulnerable because they have worked so hard in the past, which is similar to the view of the course of life [43]. They believed that their time of youth and functioning was over; an idea that also reinforced by the family and the society. They attributed their physical burnout to the overwork they had previously imposed on their bodies and likened themselves to "a brick wall waiting to fall at any moment"; this perception created feelings of misery, desperation, helplessness, and despair in them. Baluch women stopped doing strenuous activities and questioned their own functional strength Unlike middle-aged people in developed countries [44].

On the other hand, other **consequence** of the category of "transcendence" implied that with advancing age in Baluch cultural context, women's honor and respect in the eye of society and the women themselves gradually increases, which is relatively consistent with the gerotranscendence theory by Tornstamstating that (1997), "It (aging) is a shift in meta-perspective from a materialistic and rational view of the world to a more cosmic and transcendent one, normally followed by an increase in life satisfaction." [45].

**In summary**, Based on the results of this study, women believed that physical, functional, and psychological declines and diseases began with the end of fertility and menopause heralds

the aging as a reality. Therefore, women refrained from worldly activities and chose the spiritual world, because in Baluch community, as a patriarchal society [46], respect for women, increases and their social status is improved by their approaching to old age. Sergeant [11] considers menopause as an opportunity for women to refocus on their social wellbeing. Menopause seems to be a good opportunity for Baluch women, too.

This study had **limitations**. One of them was Baluch women's laconic speech, which becomes even more apparent with age. In the present study, the researcher tried to establish a closer relationship with the participants and gain their trust using their native dialect. Another limitation of this study, as all qualitative studies was its restricted generalizability to the entire community, although diversity in sampling methods could alleviate this limitation to some extent. This study is the first investigation on the impact of the event of menopause as the turning point between middle age and old age in a woman's life cycle. It should be noted that this study was conducted on a particular ethnic group; therefore, it can represent just the major challenges facing these special women.

## Conclusions

It seems that menopause plays an important role in Baluch women's view toward aging. The practical results of this study can be applied to better understand the middle-aged and older Baluch women's attitudes toward aging. Also study shows a new evolutionary and situational perspective on the lives of middle-aged menopausal Baluch women. Baluch women in menopause accept the sunset of youth and look forward to experiencing the Feeling of transcendence. Identify and respond to their needs by developing and establishing health policies to change their negative attitudes. Motivational counseling to empower these women's and change their views during this critical period is suggested.

## Supporting information

**S1 Table. Interview guid (English).**
(DOC)

**S2 Table. Interview guid (Persion).**
(DOC)

## Acknowledgments

The authors would like to thank all the women who participated in this study and an anonymous reviewer for their helpful comments.

## Author Contributions

**Conceptualization:** Shahin-Dokht Navvabi-Rigi, Farahnaz Mohammadi-Shahboulaghi, Mahshid Foroughan.

**Data curation:** Shahin-Dokht Navvabi-Rigi, Farahnaz Mohammadi-Shahboulaghi.

**Formal analysis:** Shahin-Dokht Navvabi-Rigi, Farahnaz Mohammadi-Shahboulaghi, Mahshid Foroughan.

**Investigation:** Shahin-Dokht Navvabi-Rigi.

**Methodology:** Shahin-Dokht Navvabi-Rigi, Farahnaz Mohammadi-Shahboulaghi, Mahshid Foroughan.

**Project administration:** Shahin-Dokht Navvabi-Rigi, Mahshid Foroughan.

**Resources:** Shahin-Dokht Navvabi-Rigi, Farahnaz Mohammadi-Shahboulaghi, Mahshid Foroughan.

**Supervision:** Farahnaz Mohammadi-Shahboulaghi, Mahshid Foroughan, Yadollah Abolfathi Momtaz.

**Validation:** Shahin-Dokht Navvabi-Rigi, Farahnaz Mohammadi-Shahboulaghi, Mahshid Foroughan, Yadollah Abolfathi Momtaz.

**Visualization:** Shahin-Dokht Navvabi-Rigi, Farahnaz Mohammadi-Shahboulaghi, Mahshid Foroughan.

**Writing – original draft:** Shahin-Dokht Navvabi-Rigi.

**Writing – review & editing:** Shahin-Dokht Navvabi-Rigi, Farahnaz Mohammadi-Shahboulaghi, Mahshid Foroughan, Yadollah Abolfathi Momtaz.

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
