## [Decision Letter · Decision Letter 0]

25 Aug 2022

PONE-D-21-38238Weaving a Cocoon on the Way to Aging Transcendence: Grounded Theory Study on Aging perception during Menopause TransitionPLOS ONE

Dear Dr. navvabi-rigi,

Thank you for submitting your manuscript to PLOS ONE. After careful consideration, we feel that it has merit but does not fully meet PLOS ONE’s publication criteria as it currently stands. Therefore, we invite you to submit a revised version of the manuscript that addresses the points raised during the review process.

We look forward to receiving your revised manuscript.

Kind regards,

Nülüfer Erbil, Ph.D, Prof.

Academic Editor

PLOS ONE

Journal Requirements:

2. PLOS ONE does not copy edit accepted manuscripts (https://journals.plos.org/plosone/s/criteria-for-publication#loc-5). To that effect, please ensure that your submission is free of typos and grammatical errors.

*Please include additional information regarding the interview guide used in the study and ensure that you have provided sufficient details that others could replicate the analyses. Please include a copy, in both the original language and English, as Supporting Information.

Reviewers' comments:

Reviewer's Responses to Questions

**Comments to the Author**

1. Is the manuscript technically sound, and do the data support the conclusions?

Reviewer #1: Yes

Reviewer #2: Yes

2. Has the statistical analysis been performed appropriately and rigorously? 

Reviewer #1: Yes

Reviewer #2: Yes

3. Have the authors made all data underlying the findings in their manuscript fully available?

Reviewer #1: Yes

Reviewer #2: Yes

4. Is the manuscript presented in an intelligible fashion and written in standard English?

Reviewer #1: Yes

Reviewer #2: Yes

5. Review Comments to the Author

Reviewer #1: Dear Author,

First of all thank you for submitting your review for this article. An interesting subject and a nice study. I enjoyed reading about this elegant study. Content is well written.

It can be published.

Reviewer #2: Weaving a Cocoon on the Way to Aging Transcendence: Grounded Theory Study on Aging perception during Menopause Transition

Dear author,

You can find my suggestions for your article, which includes an important and current issue, below.

• The problem description is pretty well written, but, In the inclusion criteria, isn't 60 days a short period of time to experience menopausal symptoms in terms of the perception of menopause? Is there any literature on this?

• It would be appropriate to include "having a psychiatric diagnosis" in the exclusion criteria.

• It is not clear exactly how the sample was determined, it should be detailed.

• Statistical methods used in data analysis should be written clearly. Information about content analysis should be given.

• How and by whom was content analysis evaluated?

• In which program were the records analysed?

• The writing of the Discussion section is disorganized. Adding the results of your study first, then the literature findings, and then the comments will make it more organized. Commentary sentences should be placed at the end of the paragraph.

• Reference spelling rules and punctuation should be paid attention to.

6. PLOS authors have the option to publish the peer review history of their article (what does this mean?). If published, this will include your full peer review and any attached files.

Reviewer #1: No

Reviewer #2: No

---

## [Author Response · Author response to Decision Letter 0]

15 Sep 2022

rebuttal letter 

Dear Reviewer #2:

1. In the inclusion criteria, isn't 60 days a short period of time to experience menopausal symptoms in terms of the perception of menopause? Is there any literature on this?

Response:

Participants were experiencing of the transition of menopause & post menopause and they selected Based on STRAW Staging System (1)

1) Siobán D. Harlow, Margery Gass, Janet E. Hall, Roger Lobo, Pauline Maki, Robert W. Rebar, Sherry Sherman, Patrick M. Sluss, Tobie J. de Villiers, for the STRAW + 10 Collaborative Group, Executive Summary of the Stages of Reproductive Aging Workshop + 10: Addressing the Unfinished Agenda of Staging Reproductive Aging, The Journal of Clinical Endocrinology & Metabolism, Volume 97, Issue 4, 1 April 2012, Pages 1159–1168, https://doi.org/10.1210/jc.2011-3362

2. • It would be appropriate to include "having a psychiatric diagnosis" in the exclusion criteria.

Response:

Thanks. This criteria considered in the research, thus added

3. • It is not clear exactly how the sample was determined, it should be detailed.

Response:

Add: . The first step Purposive Sampling was performed based on inclusion criteria with maximum diversity in sampling (being single or married, having children or not, diferrent menopause ages, socioeconomic status) by researcher in the comprehensive health service centers, in Zahedan in the southeast of Iran and contacted the participants and conducted interviews at the center or at home , at their request. purposefully to recruit the participants. In the next step, theoretical sampling was performed, and based on the results of data analysis the main concepts were formed, and the relationships between these concepts were explained.

4. • Statistical methods used in data analysis should be written clearly. Information about content analysis should be given.

Response:

Data analysis was performed according to Corbin and Straus’s approach (2015) in the grounded theory in four phases: (1) identifying and developing concepts; (2) analyzing data for the context; (3) entering the process stage into analysis; and (4) integrating categories to build a theory. . The statistic methods used in order to analyze and identify the data which were the context, process and consequences based on paradigm analytic tools such as asking questions from the data (sensitive questions, theoretical questions, Questions that were more practical in nature and guiding questions), and comparing (constant comparison and theoretical comparison). In this study, each interview was recorded, typed and transcribed, and initial code and then initial concept by researcher identified. In analyzing the data for context, concepts linked in order to establish between categories, and linked them together in a new concept through ongoing comparison. In the stage of process of the analysis, the relationships between intra-categories systematically and integrating the categories (subject labels based on integrated memo titles used to name concepts). In the stage of integrating the categories, the study’s storyline, core concept, hypothesis emerged by way of the relationship formed between the categories and clarification 

• How and by whom was content analysis evaluated?

Response:

Add: . The analysis was done by researcher and it was evaluated by two co-worker who were an expert in qualitative research.

5. • In which program were the records analysed?

Response:

Add: . In this Statistical analysis was not used any analytical app and it was done manually.

• The writing of the Discussion section is disorganized. Adding the results of your study first, then the literature findings, and then the comments will make it more organized. Commentary sentences should be placed at the end of the paragraph.

Response:

6. It checked &corrected

7. • Reference spelling rules and punctuation should be paid attention to.

Response: 

It checked &corrected

List of references completed & corrected : ref(8-14-16-17-19-20-21-23-24-25-30-31-32-33-34-35-36-39-40-41-42-44)

There are not any retracted articles.

---

## [Decision Letter · Decision Letter 1]

14 Oct 2022

Weaving a Cocoon on the Way to Aging Transcendence: Grounded Theory Study on Aging perception during Menopause Transition

PONE-D-21-38238R1

Dear Dr. Farahnaz Mohammadi-Shahboulaghi,

We’re pleased to inform you that your manuscript has been judged scientifically suitable for publication and will be formally accepted for publication once it meets all outstanding technical requirements.

Kind regards,

Nülüfer Erbil, Ph.D, Prof.

Academic Editor

PLOS ONE

---

## [Editor Report · Acceptance letter]

20 Oct 2022

PONE-D-21-38238R1 

Weaving a Cocoon on the Way to Aging Transcendence: Grounded Theory Study on Aging perception during Menopause Transition 

Dear Dr. Mohammadi-Shahboulaghi:

I'm pleased to inform you that your manuscript has been deemed suitable for publication in PLOS ONE. Congratulations! Your manuscript is now with our production department. 

Kind regards, 

on behalf of

Dr. Nülüfer Erbil 

Academic Editor

PLOS ONE